# High-frequency transport and zero-sound
# in an array of SYK quantum dots

Aleksey V. Lunkin[1,2⋆] and Mikhail V. Feigel'man[1]

**1** L.D.Landau Institute for Theoretical Physics, Chernogolovka, Russia
**2** HSE University, Moscow, Russia

⋆ lunkin@itp.ac.ru

## Abstract

We study an array of strongly correlated quantum dots of complex SYK type and account for the effects of quadratic terms added to the SYK Hamiltonian; both local terms and inter-dot tunneling are considered in the non-Fermi-liquid temperature range $T \gg T_{FL}$. We take into account soft-mode fluctuations and demonstrate their relevance for physical observables. Electric $\sigma(\omega, p)$ and thermal $\kappa(\omega, p)$ conductivities are calculated as functions of frequency and momentum for arbitrary values of the particle-hole asymmetry parameter $\mathcal{E}$. At low-frequencies $\omega \ll T$ we find the Lorenz ratio $L = \kappa(0,0)/T\sigma(0,0)$ to be non-universal and temperature-dependent. At $\omega \gg T$ the conductivity $\sigma(\omega, p)$ contains a pole with nearly linear dispersion $\omega \approx s p \sqrt{\ln \frac{\omega}{T}}$ reminiscent of the "zero-sound", known for Fermi-liquids. We demonstrate also that the developed approach makes it possible to understand the origin of heavy Fermi liquids with anomalously large Kadowaki-Woods ratio.

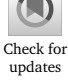

# 1  Introduction

The Sachdev-Ye-Kitaev (SYK) model [1] describes a zero-dimensional system of $N$ interacting fermions. The energy levels of electrons are degenerate, and the interaction matrix elements are random with zero mean and variance with typical energy scale $J$. On the one hand, the SYK model presents a fully analytical description of a strongly interacted fermionic system [2–5]. On the other hand, the model has "asymptotic symmetry" (at the mean-field level), which leads to the existence of Goldstone mode with strong infra-red fluctuations.

From the condensed matter theory viewpoint, two questions seem to be natural: 1) how could one extend the SYK model to describe fermionic systems in dimensions $d \geq 1$ ? 2) how the properties of this model are modified under reasonable perturbations? The simplest way to extend SYK model to higher dimensions is to consider an array of SYK quantum dots coupled by weak tunnelling [8]. Using the saddle-point approximation valid at $N \to \infty$, the authors of Ref. [8] predicted non-Fermi-liquid temperature behavior of $dc$ conductivity $\sigma(T) \propto 1/T$ [9–13] in the temperature range $T \gg T_{FL}$, while at $T$ below the crossover temperature $T_{FL}$ the usual Fermi-liquid behavior with $\sigma(T \to 0) \to$ const was found. A number of other papers have also considered similar issues [14–17].

The answer to the second question becomes non-trivial once the analysis goes beyond the mean-field limit $N \to \infty$. At finite values of $N$, quadratic perturbations of the strength $\Gamma \ll J$ added to the Hamiltonian of a single SYK quantum dot with basic energy scale $J$ leave intact [18] the non-Fermi-liquid ground-state for sufficiently small $\Gamma < \Gamma_c \approx J/N$. The behavior of the mixed $SYK_4 + SYK_2$ quantum dot at stronger perturbation magnitudes $\Gamma_c < \Gamma \leq J/\sqrt{N}$ was studied in Ref. [19] where unexpected "polaron" bound-state solution was found. Physical significance of the corresponding bound-state energy was clarified in Ref. [20], where non-equilibrium generalization of the approach [19] was developed. We demonstrated that high-frequency parametric modulation of the $SYK_2$ part of the Hamiltonian leads at $T_{FL} = \Gamma^2/J < T \ll \Gamma$ to the energy dissipation rate that is peaked at the frequency corresponding to the polaron energy $\sim \Gamma$. As a byproduct, we demonstrated that physical susceptibilities at $T_{FL} \ll T \leq \Gamma$ acquire important fluctuational contributions while single-fermion Green's function is described by pure $SYK_4$ saddle-point solution.

In the present paper we study non-equilibrium properties of an array of $SYK_4 + SYK_2$ quantum dots and extend substantially the results of Ref. [8]. On technical level, our extension amounts to the account for the effects of fluctuation of the soft reparametrization mode (absent in Ref. [8] due to the limit $N \to \infty$ employed there). Clearly, any real quantum dot may contain finite number of electrons only, and this number does not need to be very large generically, thus the account for the effects arsing due to finite $N$ is essential. Moreover, our analysis demonstrate the existence of important temperature region $T_{FL} < T < \Gamma$ where the said fluctuation effects contribute considerably to thermal conductivity even in the $N \to \infty$ limit, while static electric conductivity is not affected by those effects and is given correctly by the results of Ref. [8]. Another direction of our study where it differs from Ref. [8] and similar papers is that we consider frequency- and momentum-dependent electric and ther-

mal conductivities; moreover, we consider general situation with an arbitrary particle-hole asymmetry. Last but not least, we consider more general model where quadratic terms in the Hamiltonian are present both as tunneling terms (with strength $w$) and *inside* quantum dots, with strength $\Gamma$. It will allow us to describe the situation of "heavy Fermi liquid" with anomalously large Kadowaki-Woods ratio known for $^3$He under pressure and some HTSC materials, see [21].

In our model SYK dots are coupled by tunnelling matrix elements $t_{ij}$ with the typical energy scale $w^2 = N \overline{t_{ij}^2}$. Intra-dot quadratic terms are characterized, like in Ref. [20], by random matrix elements $\Gamma_{ij}$ with effective strength $\Gamma$. In the absence of intra-dot interactions, the system's bandwidth would be of the order of $w$, like in usual granular metal [22]. Our goal is to study the opposite situation when intra-dot interactions are very strong, with typical energy scale $J \gg w, \Gamma$. At the same time, we do not include long-range (inter-dot Coulomb) interactions into our model. For this reason, any response to electric field we calculate below should be understood as response to the local field, but not to the external one.

The role of quadratic terms of the Hamiltonian depends crucially on the temperature scale considered. At relatively high temperatures $J \gg T \geq \tilde{\Gamma}$ charge and energy transport are described by the large-$N$ saddle-point approximation developed in Ref. [8]. Here $\tilde{\Gamma} \sim \max(\Gamma, w)$ is the effective strength of quadratic perturbations, and we will assume it to be large in comparison to the critical magnitude $\Gamma_c \approx J/N$ determined in Ref. [18]. We will be most interested in the moderate-$T$ region $\tilde{\Gamma}^2/J = T_{FL} \leq T \leq \tilde{\Gamma}$ which is analogous to the region $\Gamma^2/J \leq T \leq \Gamma$ studied in Ref. [20]. It will be shown that in this temperature region transport properties are strongly affected by fluctuations of the soft reparametrization mode known to determine the behavior of SYK model [4,5]. The condition $\tilde{\Gamma} \gg \Gamma_c$ mentioned above will allow us to consider these fluctuations Gaussian. While single-Fermion Green's function is well-described by the $N \to \infty$ approximation, fluctuations crucially affect kinetic properties at $T \leq \tilde{\Gamma}$.

The method we employ is based on the use of continuity equations for charge and energy currents in presence of external sources: electric field and gradient of the Luttinger potential [23]. We show that reparametrization mode is intrinsically related to the energy current. We consider frequency- and momentum-dependent conductivities, paying major attention to the asymptotic regions of low frequencies $\omega \ll T$ and high frequencies $\omega \gg T$. In the static limit $\omega \to 0$ we confirm the results of Ref. [8] for electric conductivity $\sigma$, while thermal conductivity $\kappa$ is found to contain additional terms that become important at $T \leq \tilde{\Gamma}$. As a result, the Lorenz ratio $L = \kappa/T\sigma$ appears to be non-universal and strongly temperature-dependent. At low frequencies and low momenta, the dispersion of electric and thermal conductivities is characterized by diffusion laws. Charge diffusion constant $D_e$ scales $\propto 1/T$ like conductivity, while energy diffusion constant $D_T$ demonstrates rather nontrivial $T$-dependence with a maximum at $T \sim \tilde{\Gamma}$.

In the high-frequency region $\omega \gg T$ we found a zero-sound - like mode which provides a pole in the frequency- and momentum-dependent conductivity $\sigma(\omega, p)$ when temperature is low compared to the bandwidth, $T \ll w$. Decay rate of this mode $1/\tau(\omega) \approx \omega/\ln \frac{\omega}{T}$ is somewhat smaller than its frequency at $T \ll \omega \ll w$. Dispersion of this mode is $\omega(p) = sp\sqrt{\ln(sp/T)}$ where velocity $s \propto w$ is temperature-independent. Remarkably, neither the largest energy scale of the problem $J$, nor the number $N$ of fermions in each dot enter these results, which form a new "universality class" of strongly interacting Fermi-system. High-frequency thermal conductivity $\kappa(\omega, p)$ contains, in general, two contributions. One of them is proportional to electric conductivity $\sigma(\omega, p)$ and to the square of particle-hole asymmetry parameter $\mathcal{E}$; the second contribution is present also at $\mathcal{E} = 0$, we will call it "intrinsic" and denote as $\tilde{\kappa}(\omega, p)$ below. This contribution to thermal conductivity contains a weakly-dispersive resonance at the frequency $\omega_0 \sim \tilde{\Gamma}$, similar to the result of Ref. [20] for a single SYK dot.

The rest of the paper is organized as follows: in Sec.2 we introduce our model and discuss

the source terms needed to calculate susceptibilities. In Sec. 3 we describe mean-field solution of the model. The effective action for small fluctuations in the presence of the sources is derived in Sec. 4. Section 5 contains our main results for electric and thermal conductivities. Our conclusions are provided in Sec. 6. Some technical details are present in the Appendix.

## 2 The model

We study an array of SYK quantum dots with $N$ fermions each. The Hamiltonian is as follows:

$$H = \sum_{\mathbf{r}} \left\{ H_{\mathbf{r}} + \sum_{\delta \mathbf{r}} \sum_{i,j} \left( t_{\mathbf{r},i;\mathbf{r}+\delta\mathbf{r},j} \psi^\dagger_{\mathbf{r},i} \psi_{\mathbf{r}+\delta\mathbf{r},j} + h.c. \right) \right\},$$

$$H_{\mathbf{r}} = \frac{1}{2!} \sum_{i,j,k,l} J_{i,j;k,l;\mathbf{r}} \, \mathcal{A} \left\{ \psi^\dagger_{\mathbf{r},i} \psi^\dagger_{\mathbf{r},j} \psi_{\mathbf{r},k} \psi_{\mathbf{r},l} \right\} + \sum_{0<i,j\leq N} \Gamma_{i,j,\mathbf{r}} \psi^\dagger_{\mathbf{r},i} \psi_{\mathbf{r},j} - \mu \hat{N} \,. \tag{1}$$

Here $\psi_{\mathbf{r},i}$ is an annihilation operator in the dot with coordinate $\mathbf{r}$ on the $i$th site inside the dot. Here $\hat{N}$ is the operator of the total number of particles and $\mu$ is a chemical potential. The first term in the first line represents the Hamiltonian of an individual dot. The second term describes tunneling between neighboring dots. The dots are labeled by the index $\mathbf{r}$ and notation $\delta\mathbf{r}$ is used to label the neighbors of a dot $\mathbf{r}$. There are $N \gg 1$ positions for fermions in each dot (labeled by index $i, j, k \ldots$). The Hamiltonian $H_{\mathbf{r}}$ describes fermions in each single dot $\mathbf{r}$. The first term in the expression for $H_{\mathbf{r}}$ is the Hamiltonian of the SYK models for the complex fermions [24] whereas the second term in $H_{\mathbf{r}}$ is a perturbation which lifts the degeneracy of their spectrum. The sign $\mathcal{A}\{\ldots\}$ denotes the antisymmetrized product of operators [24]. Tensors $J \ldots$ and $\Gamma \ldots$ satisfy following symmetry properties: $J^*_{i,j;k,l;\mathbf{r}} = J^*_{l,k;j,i;\mathbf{r}}$, $\Gamma^*_{i,j,\mathbf{r}} = \Gamma_{j,i,\mathbf{r}}$ and $J_{j,i;k,l;\mathbf{r}} = J_{i,j;l,k;\mathbf{r}} = -J_{i,j;k,l;\mathbf{r}}$. Components of tensors $t_{\ldots}, J_{\ldots}, \Gamma_{\ldots}$ are independent random Gaussian variables with zero mean value and the following variances:

$$\overline{|J_{i,j;k,l;\mathbf{r}}|^2} = \frac{2J^2}{N^3} \,, \qquad \overline{\Gamma^2_{i,j,\mathbf{r}}} = \frac{\Gamma^2}{N} \,, \qquad \overline{t^2_{\mathbf{r},i;\mathbf{r}+\delta\mathbf{r},j}} = \frac{w^2}{N} \,. \tag{2}$$

We assume that the SYK term is dominant: $J \gg \Gamma, w$.

The operators of charge density and energy density are defined as follows:

$$\mathcal{Q}_{\mathbf{r}} = \sum_i \psi^\dagger_{\mathbf{r},i} \psi_{\mathbf{r},i} - \frac{N}{2} \,,$$

$$\mathcal{H}_{\mathbf{r}} = H_{\mathbf{r}} + \frac{1}{2} \sum_{\delta\mathbf{r}} \sum_{i,j} \left[ \left( t_{\mathbf{r},i;\mathbf{r}+\delta\mathbf{r},j} \psi^\dagger_{\mathbf{r},i} \psi_{\mathbf{r}+\delta\mathbf{r},j} + h.c. \right) + \mathbf{r} \mapsto \mathbf{r} - \delta\mathbf{r} \right]. \tag{3}$$

Note that the latter one contains symmetrized terms responsible for inter-dot tunneling. In thermal equilibrium, the average charge at a dot is equal to $\langle \mathcal{Q}_{\mathbf{r}} \rangle = \mathcal{Q}$.

To calculate charge and heat conductivity, we introduce the source terms that describe electric voltage and Luttinger potential. The Hamiltonian with the sources reads:

$$H_S(t) = H + \sum_{\mathbf{r}} \left( \mathcal{Q}_{\mathbf{r}} U(\mathbf{r}, t) + L(\mathbf{r}, t) \mathcal{H}_{\mathbf{r}} \right). \tag{4}$$

Our goal is to derive expressions for the charge and energy currents originating from the presence of these sources.

# 3 Around SYK solution

## 3.1 Action in terms of the Green's function $G$ and self-energy $\Sigma$

We will use path integral representation of the problem in the Keldysh form [25], since we are interested in non-equilibrium properties. To simplify discussion of the derivation to the soft-mode action we will omit terms with sources. We will restore them during the discussion of the continuity equations. The full action with sources could be found in the appendix.

The action is defined as an integral over Keldysh contour and has the form:

$$S = \int_{\mathcal{C}} \left( \sum_{\mathbf{r},i} \psi^{\dagger}_{\mathbf{r},k} i\partial_t \psi_{\mathbf{r},k} - H_S(t) \right) dt .\tag{5}$$

First of all we need to take average over disorder. Since the Keldysh functional integral is normalized to unity (in the absence of source fields), the averaging of the functional with the action (5) can be performed in the straightforward way, making use of Eqs. (2).

We would like to rewrite the action in terms of the matrix field $G(t_1, t_0|\mathbf{r})$ defined as:

$$G(t_1, t_0|\mathbf{r}) = -\frac{i}{N} \sum_l \psi_{\mathbf{r},l}(t_1)\psi^{\dagger}_{\mathbf{r},l}(t_0),\tag{6}$$

and axiliary matrix field $\Sigma(t_1, t_0|\mathbf{r})$ which plays the role of Lagrangian multiplier conjugated to $G(t_1, t_0|\mathbf{r})$. Then the action becomes quadratic in terms of fermions, and they can be integrated out. The action acquires then the following form:

$$\begin{aligned}
S = &-iN \sum_{\mathbf{r}} \int_{\mathcal{C}} dt_1 dt_0 (\Sigma(t_1, t_0|\mathbf{r}) + \Sigma_{free}(t_1, t_0)) G(t_0, t_1|\mathbf{r}) - iN \ tr \log\{\hat{\Sigma}\} \\
&+ iN \int_{\mathcal{C}} dt_1 dt_0 \sum_{\mathbf{r}} \left\{ \frac{J^2}{4} \left[ G(t_1, t_0|\mathbf{r}) G(t_0, t_1|\mathbf{r}) \right]^2 + \frac{\Gamma^2}{2} G(t_1, t_0|\mathbf{r}) G(t_0, t_1|\mathbf{r}) \right. \\
&\left. + w^2 \sum_{\delta\mathbf{r}} G(t_1, t_0|\mathbf{r}) G(t_0, t_1|\mathbf{r} + \delta\mathbf{r}) \right\} .
\end{aligned}\tag{7}$$

We introduced new notation $\Sigma_{free}(t_1, t_0) \equiv \left\{ i\partial_{t_1} + \mu \right\} \delta(t_1 - t_0)$. Since $N \gg 1$ we can start from the mean-field (saddle-point) solution and then add fluctuation effects.

## 3.2 Pure SYK solution

Saddle-point equations for $\Sigma$ and $G$ can be obtained by minimization of the action (9) over these matrix variables in the absence of sources. The terms describing quadratic perturbations to the Hamiltonian are irrelevant at this stage as long as we consider temperatures $T \gg T_{FL}$, where $T_{FL} \equiv \max\{\frac{w^2}{J}, \frac{\Gamma^2}{J}\}$. It will be convenient to replace time variable $t$ by its dimensionless analog $u \equiv 2\pi Tt$. Then self-energy and Green's function will change according to

$$G(t_1, t_0|\mathbf{r}) \equiv \left( \frac{2\pi T}{J} \right)^{1/2} G(u_1, u_0|\mathbf{r}), \qquad \Sigma(t_1, t_0|\mathbf{r}) \equiv J^2 \left( \frac{2\pi T}{J} \right)^{3/2} \Sigma(u_1, u_0|\mathbf{r}).\tag{8}$$

Using these new variables we write the SYK action in the form:

$$S_{SYK} = -iN \, tr \log\{\hat{\Sigma}\} - iN \sum_{\mathbf{r}} \sum_{\sigma_1,\sigma_0} \sigma_1 \sigma_0 \int du_1 du_0 \Sigma_{free,\sigma_1\sigma_0}(u_1,u_0) G_{\sigma_1\sigma_0}(u_0,u_1|\mathbf{r})$$

$$+ iN \sum_{\sigma_1,\sigma_0} \sigma_1 \sigma_0 \int du_1 du_0 \sum_{\mathbf{r}} \Big\{ \frac{1}{4} \big[ G_{\sigma_1\sigma_0}(u_1,u_0|\mathbf{r}) G_{\sigma_0\sigma_1}(u_0,u_1|\mathbf{r}) \big]^2 \tag{9}$$

$$- \Sigma_{\sigma_1\sigma_0}(u_1,u_0|\mathbf{r}) G_{\sigma_0\sigma_1}(u_0,u_1|\mathbf{r}) \Big\}.$$

Here we split a Keldysh contour $\mathcal{C}$ into positive and negative branches [25], so that summation variables $\sigma_{0,1}$ take values $\pm1$. Minimization of the action (12) provides the saddle-point functional equations in the form:

$$\Sigma_{\sigma_1\sigma_0}(u_1,u_0|\mathbf{r}) = \big[ G_{\sigma_1\sigma_0}(u_1,u_0|\mathbf{r}) G_{\sigma_0\sigma_1}(u_0,u_1|\mathbf{r}) \big]^{\frac{q}{2}-1} G_{\sigma_0\sigma_1}(u_0,u_1|\mathbf{r}),$$

$$\sum_{\sigma} \sigma \sigma_1 \int \Sigma_{\sigma_1\sigma}(u_1,u|\mathbf{r}) G_{\sigma\sigma_0}(u,u_0|\mathbf{r}) du = -\delta_{\sigma_1,\sigma_0} \delta(u_1-u_0). \tag{10}$$

We omit the term $\Sigma_{free}$ from the saddle-point equations (10) since it is parametrically small at time scales $t \gg 1/J$ where we will consider these equations. We introduce parameter $q$ which is equal to 4 in our case; this form can be important in the context of generalized SYK model (with arbitrary number of fermions, [4]) we also used this parameter for the dimensional regularization (see comment in 6) Translationally invariant solution of these equations is (see also [26]):

$$G^{(s)}_{\sigma_1\sigma_0}(u_1,u_0|\mathbf{r}) = b^\Delta g_{\sigma_1\sigma_0}(u_1-u_0), \qquad b = \frac{(1-2\Delta)\sin(2\pi\Delta)}{2\pi(\cos(2\pi\Delta)+\cosh(2\pi\mathcal{E}))}, \tag{11}$$

where we have used function $g$ defined as follows:

$$g_{\sigma_1\sigma_0}(u) \equiv ie^{-i\mathcal{E}u} \left( 4\sinh^2\left(\frac{u}{2}\right) \right)^{-\Delta} \left[ \theta(u) \begin{pmatrix} -e^{-i\pi(\Delta+i\mathcal{E})} & e^{i\pi(\Delta+i\mathcal{E})} \\ -e^{-i\pi(\Delta+i\mathcal{E})} & e^{i\pi(\Delta+i\mathcal{E})} \end{pmatrix} + \theta(-u) \begin{pmatrix} e^{-i\pi(\Delta-i\mathcal{E})} & e^{-i\pi(\Delta-i\mathcal{E})} \\ -e^{i\pi(\Delta-i\mathcal{E})} & -e^{i\pi(\Delta-i\mathcal{E})} \end{pmatrix} \right]_{\sigma_1\sigma_0}. \tag{12}$$

Here $\Delta = \frac{1}{q}$ and $\mathcal{E} \in (-\infty,\infty)$ is the parameter of particle-hole asymmetry which can be related (see [24,26]) to the average charge of the dot $\mathcal{Q}$ as follows:

$$\mathcal{Q} = -\frac{\theta}{\pi} - \left(\frac{1}{2}-\Delta\right) \frac{\sin(2\theta)}{\sin(2\pi\Delta)}, \qquad e^{2\pi\mathcal{E}} \equiv \frac{\sin(\pi\Delta+\theta)}{\sin(\pi\Delta-\theta)}. \tag{13}$$

Actually the asymmetry parameter $\mathcal{E}$ cannot be too large in its absolute value. The stable solutions of the type provided by Eq.(11) can be obtained for $|\mathcal{E}| \leq 0.14$ only, as it was found numerically in Refs. [27,28]. Although these papers studied models similar but not identical to ours, they worked with the same saddle-point equations (10), so we expect that their results for maximum value of $|\mathcal{E}|$ apply to our case as well.

### 3.3 Selection of the soft-mode manifold

The saddle-point solution (11) breaks down large symmetry of the saddle-point equations (10). Namely, symmetry transformations allowed for Eqs. (10) are as follows: for any solution $G_{\sigma_1,\sigma_0}(u_1,u_0|\mathbf{r})$ we can produce a new solution $\tilde{G}$ defined as

$$\tilde{G}_{\sigma_1,\sigma_0}(u_1,u_0|\mathbf{r}) = \big[ F'_{\sigma_1}(u_1,\mathbf{r}) F'_{\sigma_0}(u_0,\mathbf{r}) \big]^\Delta e^{i\left(\varphi_{\sigma_1}(u_1,\mathbf{r})-\varphi_{\sigma_0}(u_0,\mathbf{r})\right)} \tag{14}$$

$$\times G_{\sigma_1,\sigma_0}(F_{\sigma_1}(u_1,\mathbf{r}), F_{\sigma_0}(u_0,\mathbf{r})|\mathbf{r}).$$

Here $F(u)$ is a monotonous function, and $\varphi(u)$ is an arbitrary function. On the other hand, the solution (11) allows for much smaller symmetry group $SL(2,R)$. As a result, approximate action (12) has a zero mode. Once previously neglected term with $\Sigma_{free}$ is put back into the action, this zero mode transforms into soft Goldstone mode, like it happens in the $\sigma$-model [29]. The corresponding manifold is parametrized by the functions $F(u)$ and $\varphi(u)$ by means of transformations (36) applied to the mean-field solution (11). As a result, we obtain the following action for the soft modes:

$$S_{soft} = \int d\mathbf{r} S_{SYK,\mathbf{r}}^{(F,\phi)} + (S - S_{SYK})|_{G=G^{(F,\phi)}}, \qquad G_{\sigma_1\sigma_0}^{F,\phi}(u_1,u_0|\mathbf{r}) = b^\Delta g_{\sigma_1\sigma_0}^{(F,\phi)}(u_1,u_0|\mathbf{r}),$$

$$g_{\sigma_1\sigma_0}^{(F,\phi)}(u_1,u_0|\mathbf{r}) = \left[ F_{\sigma_1}'(u_1,\mathbf{r}) F_{\sigma_0}'(u_0,\mathbf{r}) \right]^\Delta e^{i\left( \varphi_{\sigma_1}(u_1,\mathbf{r}) - \varphi_{\sigma_0}(u_0,\mathbf{r}) \right)} \tag{15}$$

$$\times g_{\sigma_1,\sigma_0}(F_{\sigma_1}(u_1,\mathbf{r}), F_{\sigma_0}(u_0,\mathbf{r})).$$

Here $S_{SYK,\mathbf{r}}^{(F,\phi)}$ is the original action for soft modes in the SYK model which has the form:

$$S_{SYK,\mathbf{r}}^{(F,\phi)} = \sum_\sigma \sigma \int du \left\{ -C_E Sch\left( e^{F_\sigma(u,\mathbf{r})}, u \right) + C_Q \left( \varphi_\sigma'(u,\mathbf{r}) + \mathcal{E} F_\sigma'(u,\mathbf{r}) - \mathcal{E} \right)^2 \right\}. \tag{16}$$

Here

$$C_E = N\alpha_s \frac{2\pi T}{JV_0}, \qquad C_Q = NK \frac{2\pi T}{JV_0}, \tag{17}$$

where $\alpha_s \approx 0.05$ and $K \approx 1.04$ see [4,24] and $V_0$ is the system volume per single dot . The second term in $S_{soft}$ comes from quadratic perturbations in the Hamiltonian and contains terms $\propto \Gamma^2$ and $w^2$.

Soft fluctuations play a crucial role in the theory of the SYK model: they change asymptotic behaviour of the Green function, as shown in Ref. [3]). On the other hand, quadratic perturbations (terms with $\Gamma$ and $w$) partially suppress these fluctuations, as demonstrated in Ref. [19]. These fluctuations are also responsible for the kinetic properties of the system [8,20]. Bellow we will consider a quadratic action for these fluctuations to find saddle-point solution in the presence of sources. We will discuss physical properties of these fluctuations in section 5.

## 4 Quadratic action for soft modes in presence of source fields

We will consider soft-mode fluctuations around the mean-field solution (11) at Gaussian level in this section; smallness of these fluctuations is controlled by the following inequalities:

$$F_\sigma(u,\mathbf{r}) = u + f_\sigma(u,\mathbf{r}), \qquad f_\sigma'(u,\mathbf{r}) \ll 1, \qquad \varphi_\sigma'(u,\mathbf{r}) \ll 1. \tag{18}$$

We will expand action (16) up to the second order in small fluctuations $f$ and $\varphi$ (similar procedure was used in Ref. [20]).

The result can be conveniently written in the Fourier representation:

$$S_{\text{final}} = \frac{1}{2} \int \frac{d^d\mathbf{p}d\Omega}{(2\pi)^{d+1}} \left( \hat\varphi_{\mathbf{p},\Omega} + \mathcal{E}\hat{f}_{\mathbf{p},\Omega} \right)^\dagger \left\{ \left[ \hat{\mathcal{G}}_0^{(\varphi)}(\Omega) \right]^{-1} - \hat\Sigma_t^{(\varphi)}(\Omega,\mathbf{p}) \right\} \left( \hat\varphi_{\mathbf{p},\Omega} + \mathcal{E}\hat{f}_{\mathbf{p},\Omega} \right)$$

$$+ \frac{1}{2} \int \frac{d^d\mathbf{p}d\Omega}{(2\pi)^{d+1}} \hat{f}_{\mathbf{p},\Omega}^\dagger \left\{ \left[ \hat{\mathcal{G}}_0^{(f)}(\Omega) \right]^{-1} - \hat\Sigma^{(f)}(\Omega) - \hat\Sigma_w^{(f)}(\Omega,\mathbf{p}) \right\} \hat{f}_{\mathbf{p},\Omega}. \tag{19}$$

Here $d$ is the space dimension of our problem. We introduced vectors that describe fields and sources on positive ($\varphi_{\mathbf{p},\Omega}^+$) and negative ($\varphi_{\mathbf{p},\Omega}^-$) parts of the Keldysh contour. We also performed Keldysh rotation defined as follows:

$$\varphi_{\mathbf{p},\Omega}^+ = \varphi_{\mathbf{p},\Omega}^{(cl)} + \frac{\varphi_{\mathbf{p},\Omega}^{(q)}}{2}, \qquad \varphi_{\mathbf{p},\Omega}^- = \varphi_{\mathbf{p},\Omega}^{(cl)} - \frac{\varphi_{\mathbf{p},\Omega}^{(q)}}{2}. \tag{20}$$

The same notations are used for other fields and sources. $2 \times 2$ Keldysh space represented in terms of $(cl, q)$ variables is convenient for representations of matrices $\hat{\Sigma}$ and $\left[\hat{\mathcal{G}}_0^{(\varphi)}(\Omega)\right]^{-1}$. Since we consider fluctuations in Gaussian approximation only, it will be sufficient to study the behavior of retarded (R) part of Green function and self-energy.

The corresponding explicit expressions are provided below:

$$\left[\hat{\mathcal{G}}_0^{(\varphi)}(\Omega)\right]^{-1}_{(R)} = \left[\mathcal{G}_{0,(R)}^{(\varphi)}(\Omega)\right]^{-1} = C_Q \Omega^2, \qquad \left[\mathcal{G}_{0,(R)}^{(f)}(\Omega)\right]^{-1} = C_E \Omega^2(\Omega^2 + 1),$$

$$\Sigma_{(R)}^{(f)}(\Omega) = \frac{C_\Gamma}{2} \Omega^2 \psi(\Omega), \qquad \Sigma_{w,(R)}^{(\varphi)}(\Omega, \mathbf{p}) = 8 C_w \sum_{\delta \mathbf{r}} \left(\frac{\mathbf{p}\delta\mathbf{r}}{2}\right)^2 (\psi(\Omega) + 2), \qquad (21)$$

$$\Sigma_{w,(R)}^{(f)}(\Omega, \mathbf{p}) = \frac{C_w}{2} \sum_{\delta \mathbf{r}} \left\{ \Omega^2 \psi(\Omega) + \frac{1}{2}\left(\frac{\mathbf{p}\delta\mathbf{r}}{2}\right)^2 \left(2(1 + \Omega^2) + (1 + 2\Omega^2)\psi(\Omega)\right) \right\}.$$

Here we introduced the functions $\psi(\Omega)$, and constants $C_\Gamma$, $C_w$, which are defined as follows:

$$\psi(\Omega) = \Psi\left(\frac{1}{2} - i\Omega\right) - \Psi\left(-\frac{1}{2}\right), \qquad \Psi(z) = \ln'(\Gamma(z)),$$

$$C_\Gamma = \frac{N}{4\pi V_0} \frac{\Gamma^2}{JT} \sqrt{b}, \qquad C_w = \frac{N}{2\pi V_0} \frac{w^2}{JT} \sqrt{b}. \qquad (22)$$

The lattice will be considered as a generalized cubic one, so we introduce the following notations:

$$\tilde{C}_w \equiv C_w \sum_{\delta \mathbf{r}} 1 = 2d C_w, \qquad \tilde{\sigma} \equiv C_w \frac{1}{p^2} \sum_{\delta \mathbf{r}} \left(\frac{\mathbf{p}\delta\mathbf{r}}{2}\right)^2 = C_w \frac{a^2}{2}. \qquad (23)$$

Here $d$ is a dimension of the lattice and $a$ is a lattice constant; the above formulas are valid at $ap \ll 1$.

The expressions (19) and (21) for quadratic part of the soft-mode action in presence of sources is the main result of this Section. These formulae will be used below to determine kinetic properties of the system in both limits of low ($\omega \ll T$) and high ($\omega \gg T$) frequencies.

# 5 Electric and thermal transport at arbitrary frequencies

## 5.1 Noether's theorem

As we are interested in the kinetics of the system we would like to obtain expressions for currents of energy and charge in our system. We need to consider the following change of the initial fermionic fields to find these currents, according to Noether's theorem:

$$\psi_{\mathbf{r},j}(t) \mapsto e^{i\delta\varphi(t,\mathbf{r})} \psi_{\mathbf{r},j}(t + \delta f(t,\mathbf{r})). \qquad (24)$$

On the other hand, we use fields $f$ and $\varphi$ instead of fermionic ones. To connect them let us write a transformation law for the field $G$ using its definition (6):

$$G_{\sigma_1,\sigma_2}(t_1, t_2|\mathbf{r}) \mapsto e^{i(\delta\varphi_{\sigma_1}(t_1,\mathbf{r}) - \delta\varphi_{\sigma_2}(t_2,\mathbf{r}))} G_{\sigma_1,\sigma_2}(t_1 + \delta f_{\sigma_1}(t_1,\mathbf{r}), t_2 + \delta f_{\sigma_2}(t_2,\mathbf{r})|\mathbf{r}). \qquad (25)$$

This map is not equal to the symmetry map described in eq. (36). As we want to limit ourselves by soft mode fluctuations only we need to project the result of the symmetry application to the manifold of soft modes. Finally, we will obtain the following transformation law for fields $f$ and $\varphi$:

$$\varphi^\sigma(u, \mathbf{r}) \mapsto \varphi^\sigma(u, \mathbf{r}) + \delta\varphi_\sigma(u, \mathbf{r}), \qquad f^\sigma(u, \mathbf{r}) \mapsto f^\sigma(u, \mathbf{r}) + 2\pi T \delta f_\sigma(u, \mathbf{r}). \qquad (26)$$

This transformation rule reveals the physical origin of fields $\varphi$ and $f$. These fields are conjugated to the charge density fluctuations and energy density fluctuation respectively. In other words they are related as:

$$\delta Q(t, \mathbf{r}) = -\frac{\delta S}{\delta \partial_u \varphi^{(q)}(u, \mathbf{r})}, \qquad \delta E(t, \mathbf{r}) = -\frac{1}{2\pi T} \frac{\delta S}{\delta \partial_u f^{(q)}(u, \mathbf{r})}. \tag{27}$$

We can understand the meaning of coefficients in the action using this expression. Let us slightly change the temperature and chemical potential of the system. As a result, the saddle-point solution will change. Green function of the new system could be obtained from the Green function of the original system using symmetry transformation (36). For this transformation: $f^{(cl)}(u, \mathbf{r}) = \frac{\delta T}{T} u$ and $\varphi^{(q)}(u, \mathbf{r}) = \frac{\delta \mu}{2\pi T} u$. As a result, the charge density and energy density will change as:

$$\delta Q = \frac{C_Q}{2\pi T} \left( \delta \mu + 2\pi \mathcal{E} \delta T \right), \qquad \delta E = 2\pi T \left( \left[ C_E + C_\Gamma + \tilde{C}_w \right] \frac{\delta T}{T} + \mathcal{E} \delta Q \right). \tag{28}$$

Using these equations, we can note that heat capacity of the system is:

$$\left( \frac{\delta E}{\delta T} \right)_Q = 2\pi (C_E + C_\Gamma + \tilde{C}_w). \tag{29}$$

In the absence of quadratic terms, this result agrees with the heat capacity of the SYK model $C_{SYK} = 2\pi C_E$ [4]. Quadratic terms become important for heat capacity at low temperatures when

$$C_E \leq C_\Gamma + \tilde{C}_w \quad \Leftrightarrow \quad T \leq \tilde{\Gamma} = \sqrt{\Gamma^2 + 4dw^2}. \tag{30}$$

As we see, quadratic terms play the crucial role here, even at $T \gg T_{FL}$.

We also can find compressibility of the system (see also [24]):

$$\left( \frac{\delta Q}{\delta \mu} \right)_T = \frac{C_Q}{2\pi T}. \tag{31}$$

Finally, we can verify Maxwell's relation (see [17, 24, 26]):

$$\left( \frac{\delta S}{\delta Q} \right)_T = -\left( \frac{\delta \mu}{\delta T} \right)_{\mathcal{Q}} = 2\pi \mathcal{E}. \tag{32}$$

Here we have connected entropy and energy as $\delta E = T \delta S$.

In this subsection, we have considered the statistical mechanics of the system. These properties were found as a response of our system to static perturbation. In the following subsection we will find kinetic properties of the system as a response on dynamical sources.

## 5.2 Continuity equations

In the limit $N \gg 1$ we can use the saddle point equation. In the absence of the quantum component of sources the quantum component of fields at saddle-point configuration will be zero. So we have only two saddle-point equation:

$$\frac{\delta S}{\delta \varphi^{(q)}(u, \mathbf{r})} = 0, \qquad \frac{\delta S}{\delta f^{(q)}(u, \mathbf{r})} = 0. \tag{33}$$

As fields $\varphi$ and $f$ are conjugated to charge and energy densities, we conclude that the above equations are continuity equations for charge and energy densities respectively.

We can also obtain expressions for the densities of charge current and energy current as:

$$j_{\mathbf{p},\omega}^{(Q),\alpha} = -\frac{\delta S_{\text{final}}}{\delta (ip^\alpha f_{\mathbf{p},\Omega}^{(q)})^\dagger}, \qquad \text{and} \qquad j_{\mathbf{p},\omega}^{(E),\alpha} = -\frac{\delta S_{\text{final}}}{\delta (ip^\alpha \varphi_{\mathbf{p},\Omega}^{(q)})^\dagger}. \qquad (34)$$

There are non-zero currents in the system at the presence of sources ($L$ and $U$). We can solve saddle-point equations (33) to find fields $\varphi$ and $f$ in the presence of sources. Using this result we can connect currents and sources as:

$$\begin{pmatrix} j_{\mathbf{p},\omega}^{(Q),\alpha} \\ j_{\mathbf{p},\omega}^{(E),\alpha} \end{pmatrix} = \begin{pmatrix} \sigma(\mathbf{p},\omega)e^2 & 2\pi\mathcal{E}e\sigma(\mathbf{p},\omega) \\ 2\pi\mathcal{E}eT\sigma(\mathbf{p},\omega) & \tilde{\kappa}(\mathbf{p},\omega)+(2\pi\mathcal{E})^2 T\sigma(\mathbf{p},\omega) \end{pmatrix} \begin{pmatrix} -ip^\alpha U_{\mathbf{p},\omega} \\ -ip^\alpha T L_{\mathbf{p},\omega}^{(cl)} \end{pmatrix}. \qquad (35)$$

Here $e$ is the charge of electron. We have also restored the dimensionalities of currents and sources. Left upper block of the matrix (35) shows that electric conductivity at arbitrary frequency and momentum is given by $e^2\sigma(\omega,p)$, where

$$\sigma(\mathbf{p},\omega=2\pi T\Omega) = \frac{8\tilde{\sigma}\tilde{\psi}(\Omega)C_Q}{-iC_Q\Omega + 8\tilde{\sigma}p^2\frac{\tilde{\psi}(\Omega)}{-i\Omega}}, \qquad \text{where} \quad \tilde{\psi}(\Omega) = \psi(\Omega) + 2 \approx -i\Omega, \qquad (36)$$

$$\text{at} \quad \Omega \to 0. \qquad (37)$$

Recall that notations $C_Q$ and $\tilde{\sigma}$ are defined in Eqs. (17) and (23) correspondingly.

The expression for heat conductivity (right-bottom matrix element in Eq. (35)) contains two terms. The first one describes thermal current without transport of charge (we call it intrinsic thermal conductivity):

$$\tilde{\kappa}(\mathbf{p},\omega=2\pi T\Omega) = \tilde{\sigma}T\pi^2 \qquad (38)$$
$$\times \frac{\left[C_E(\Omega^2+1) - \tilde{C}\psi(\Omega)\right]\left((\Omega^2+1)\tilde{\psi}(\Omega) + 5\Omega^2\psi(\Omega)\right) + 2\Omega^2\psi^2(\Omega)C_\Gamma}{-iC_E\Omega(\Omega^2+1) + \frac{\tilde{C}}{2}i\Omega\psi(\Omega) + \frac{\tilde{\sigma}}{4}p^2\left((\Omega^2+1)\frac{\tilde{\psi}(\Omega)}{-i\Omega} + i\Omega\psi(\Omega)\right)}.$$

Here we introduced a new constant $\tilde{C}$, see Eqs. (22) and (23):

$$\tilde{C} \equiv C_\Gamma + \tilde{C}_w = \frac{N\sqrt{b}}{4\pi V_0 J T}\tilde{\Gamma}^2. \qquad (39)$$

This constant describes contribution of perturbations to the heat capacity (see also (28)).

The second term in the right-bottom block of (35) is due to charge transport; the total thermal conductivity is given then by

$$\kappa(\mathbf{p},\omega) = \tilde{\kappa}(\mathbf{p},\omega) + (2\pi\mathcal{E})^2 T\sigma(\mathbf{p},\omega). \qquad (40)$$

Finally, off-diagonal elements of the conductivity matrix (35) define a Seebeck ($\mathscr{S}$) coefficient, which is equal to

$$\mathscr{S} = \frac{2\pi\mathcal{E}}{e} = \frac{\partial S}{\partial Q}, \qquad (41)$$

where $S$ is the entropy of the system. This result was obtained in Refs. [24, 26].

Below we consider simplified expressions for electric and thermal conductivities at low frequencies $\omega \ll T$ and and high frequencies $\omega \gg T$.

## 5.3 Hydrodynamic limit

In the hydrodynamic limit $\omega \ll T$, the system is in a local equilibrium. In this limit, electric conductivity and intrinsic heat conductivity acquire the following form:

$$\sigma(\mathbf{p},\omega) = (2\pi)^2 \tilde{\sigma} \frac{-i\omega \frac{C_Q}{2\pi T}}{-i\omega \frac{C_Q}{2\pi T} + (2\pi)^2 \tilde{\sigma} p^2} \,, \tag{42}$$

$$\tilde{\kappa}(\mathbf{p},\Omega) = \frac{\tilde{\sigma} T \pi^4}{2} \frac{-i2\pi \left[ C_E + 2\tilde{C} \right] \omega}{-i2\pi \left[ C_E + \tilde{C} \right] \omega + \frac{\tilde{\sigma} T \pi^4}{2} p^2} \,. \tag{43}$$

Uniform static electric conductivity is equal to

$$\sigma_0(T) = \sigma(\mathbf{0}, \Omega \to 0; T) = \pi \sqrt{b} \frac{N}{V_0} \frac{w^2}{J T} a^2 \,. \tag{44}$$

Dispersions of both conductivities as functions of frequency and momentum are characterized by diffusive dependencies. The diffusion coefficients are as follows (see also (29, 31)):

$$D_e = \sigma_0(T) \left[ \frac{\delta Q}{\delta \mu} \right]^{-1} \propto \frac{1}{T} \,, \qquad \text{charge diffusion}\,, \tag{45}$$

$$D_T = \frac{\pi^2}{8} \sigma_0(T) \, T \left[ \frac{\delta E}{\delta T} \right]^{-1} \propto \begin{cases} T & \text{at } T \ll \tilde{\Gamma} \\ T^{-1} & \text{at } T \gg \tilde{\Gamma} \end{cases} \,, \qquad \text{energy diffusion}\,. \tag{46}$$

Lorenz ratio $L$ in the zero-$\omega$ and zero-$p$ limit is given (see Eqs. (43), (40), (17) and (23) ) by

$$L = \frac{\kappa(\mathbf{0},0)}{T \sigma(\mathbf{0},0)} + (2\pi\mathcal{E})^2 = \frac{\pi^2}{8} \frac{C_E + 2\tilde{C}}{C_E + \tilde{C}} + (2\pi\mathcal{E})^2 = \frac{\pi^2}{8} \frac{\alpha_s + \sqrt{b} \frac{\tilde{\Gamma}^2}{(2\pi T)^2}}{\alpha_s + \frac{\sqrt{b}}{2} \frac{\tilde{\Gamma}^2}{(2\pi T)^2}} + (2\pi\mathcal{E})^2 \,. \tag{47}$$

At relatively high temperatures $T \geq \tilde{\Gamma}$ the heat capacity is dominated by pure SYK term, $C_E \gg C_\Gamma$, and Lorenz ratio follows the result of Ref. [17], $L = \pi^2/8 + (2\pi\mathcal{E})^2$ . However, at lower temperatures $T \ll \tilde{\Gamma}$ the heat capacity is determined by the term coming from quadratic perturbation in the Hamiltonian, $\tilde{C} \gg C_E$, and we find for Lorenz ratio another result:

$$L = \frac{\pi^2}{4} + (2\pi\mathcal{E})^2 \,. \tag{48}$$

We emphasize that considerable variation of the Lorenz ratio with temperature occurs in domain $T \sim \tilde{\Gamma} \gg T_{FL}$ where modifications of the saddle-point solution are small. The whole effect upon $L(T)$ which we found, comes due to soft-mode fluctuations. At lowest temperatures $T \leq T_{FL}$ the Lorenz ratio returns to its Fermi-liquid value $L_{FL} = \frac{\pi^2}{3}$ irrespectively of $\mathcal{E}$.

## 5.4 High-frequency, low-$T$ limit

Relaxation rate in our system is bounded as $1/\tau_{\text{inel}} \leq T$ thus the high-frequency range $\omega \gg T$ corresponds to its non-equilibrium behaviour. Surprisingly, we find electric current response containing a pole in the complex plane of frequency with nearly linear dispersion and relatively small decay rate. This resonance mode looks similar to the zero sound in Fermi-liquids [30] in spite of completely different nature of our system. Electric conductivity in this regime can be written as follows (see Eq.(37) ):

$$\sigma(\mathbf{p},\omega) = 8\tilde{\sigma} C_Q \frac{i \frac{\omega}{2\pi T} \psi(\frac{\omega}{2\pi T})}{C_Q \left[ \frac{\omega}{2\pi T} \right]^2 - 8\tilde{\sigma} p^2 \psi(\frac{\omega}{2\pi T})} \,, \qquad \psi(\Omega) \approx \ln(|\Omega|) - i\frac{\pi}{2} \text{sign}(\Omega) \,. \tag{49}$$

At $\Omega \gg 1$ the real part of the function $\psi(\Omega)$ is much larger than its imaginary part which makes possible propagation of waves. The dispersion of waves is determined by the position of the pole in Eq.(49):

$$\omega^2(p) = s^2 p^2 \ln\left(\frac{sp}{T}\right), \qquad s = 2\sqrt{D_e T}. \tag{50}$$

The characteristic velocity $s$ is temperature-independent as $D_e \sim \frac{1}{T}$ and we can express it as:

$$s = \frac{2b^{1/4}}{K^{1/2}} aw. \tag{51}$$

For the condition $sp \gg T$ to be satisfied, the temperature should be low enough, $T \ll w$. Note that $s$ does not depend on the largest energy scale $J$, although the mere existence of this mode is related to strong electron-electron interaction. Similar situation was found in our previous study [20] of the energy absorption in a single $SYK_4 + SYK_2$ quantum dot. Life-time $\tau(p)$ of the "zero-sound" modes (50) is relatively long at $\omega \gg T$, namely: $\omega(p)\tau(p) \approx \ln\frac{\omega}{T}$.

Thermal conductivity in the high-frequency regime contains two terms, one of them is proportional to $\sigma(\mathbf{p}, \omega)$ given by Eq.(49), see general relation (40). The second term, the intrinsic thermal conductivity, is given by

$$\tilde{\kappa}(\mathbf{p}, \omega) = 3\tilde{\sigma}\pi\psi\left(\frac{\omega}{2\pi T}\right) \frac{i\omega\left[C_E\left[\frac{\omega}{2\pi T}\right]^2 - (\tilde{C} - \frac{1}{3}C_\Gamma)\psi(\frac{\omega}{2\pi T})\right]}{C_E\left[\frac{\omega}{2\pi T}\right]^2 - \frac{\tilde{C}}{2}\psi(\frac{\omega}{2\pi T}) - \frac{\tilde{\sigma}}{2}p^2\psi(\frac{\omega}{2\pi T})}. \tag{52}$$

As a function of frequency, $\tilde{\kappa}(\mathbf{p}, \omega)$ contains a pole at the frequency $\omega_{th}(p)$ which is close to a single resonance frequency $\omega_0$ defined below, since we always have the condition $\tilde{\sigma}p^2 \ll C_E$ fulfilled. The equation for $\omega_0$ reads

$$\left[\frac{\omega_0}{2\pi T}\right]^2 = \frac{\tilde{C}}{2C_E}\ln\left(\frac{\omega_0}{2\pi T}\right) \quad \rightarrow \quad \omega_0 = \frac{b^{1/4}}{2\sqrt{\alpha_S}}\left[\tilde{\Gamma}^2\ln\frac{\omega_0}{2\pi T}\right]^{1/2}. \tag{53}$$

To characterize decay rate and dispersion of the intrinsic thermal mode $\omega_{th}(p)$ we set $\omega_{th}(p) \equiv sign(\omega)\omega_0 + \delta\omega_{th}(p)$ and find:

$$\delta\omega_{th}(p) = -\frac{i\pi}{4}\frac{\omega_0}{\ln\left(\frac{\omega_0}{2\pi T}\right)} + \frac{sign(\omega)\sigma_0\omega_0}{8\pi^2 C}p^2, \tag{54}$$

where $C = C_E + \tilde{C}$. The mode $\omega_{th}(p)$ is completely analogous to the frequency of resonant absorption found in Ref. [20].

# 6 Conclusions

We have studied a model of strongly correlated electron liquid described by $SYK_4 + SYK_2$ quantum dots coupled by single-particle tunneling terms. Effective strength of quadratic perturbations $\tilde{\Gamma} = \sqrt{\Gamma^2 + 4dw^2}$ is small compared to interaction strength $J$, but it is still much larger than the critical value $\Gamma_c \approx J/N$ found in Refs. [18, 19]. Therefore non-Fermi-liquid behaviour of the system is limited from below by the temperature scale $T_{FL} \approx \tilde{\Gamma}^2/J$, and we consider the range of temperatures $T \geq T_{FL}$ only. Our results do not depend on the number $N$ of electrons in each quantum dot, as long as we work at $\tilde{\Gamma} \gg \Gamma_c$. However, at smaller $\tilde{\Gamma}$ the parameter $N$ becomes important; this region of parameters is still to be studied.

The generic case of an electron - hole asymmetric model, $\mathcal{E} \neq 0$, is considered. General expressions for frequency-dependent and momentum-dependent electric and thermal conductivities $\sigma(\mathbf{p}, \omega)$ and $\kappa(\mathbf{p}, \omega)$ are derived. For non-zero asymmetry parameter $\mathcal{E}$, thermal conductivity contains both the "intrinsic" (unrelated to charge transport) term, and the term proportional to electric conductivity.

Qualitatively new results are obtained for the intermediate temperature interval $T_{FL} \ll T \leq \tilde{\Gamma}$ where fluctuations of the reparametrization soft mode make strong effect upon kinetic properties. Physically this mode can be understood as space-time fluctuations of the local energy density in strongly interacting system. It is thus natural that these fluctuations affect thermal conductivity, which is strongly modified already in the static $dc$ limit, studied previously within saddle-point approximation [8]. Namely, the Lorenz ratio $L = \kappa/T\sigma$ is found to be temperature - dependent, it also depends on particle-hole asymmetry parameter $\mathcal{E}$, see Eqs. (47,48). Electric conductivity in the $dc$ limit is not sensitive to soft-mode fluctuations and demonstrate "strange-metal" scaling $\sigma \propto 1/T$ in agreement with Ref. [8], as well as kinetic relaxation time $\tau(T) \sim 1/T$.

Soft-mode fluctuations become even more important for the high-frequency $\omega \gg T$ transport that is strongly non-equilibrium since $\omega\tau(T) \gg 1$. We found that high-frequency electric conductivity contains a pole contribution (49), with the dispersion relation of nearly-linear form, Eq.(50), which seems to be a "strongly - correlated partner" of a zero-sound mode known for Fermi-liquids. This mode exists in an intermediate temperature range $T_{FL} < T \ll w$. We emphasize that "sound velocity" given by Eq.(51) does not depend on interaction parameter $J$, although existence of such a mode is related to strong interactions. Lifetime of these nearly-sound modes $\tau_s(p)$ is relatively long in the $\omega/T \gg 1$ limit: $\tau_s(p)\omega(p) \approx \ln\frac{sp}{T}$.

High-frequency thermal conductivity contains (at $\mathcal{E} \neq 0$) similar pole contribution; in addition, the intrinsic contribution to high-frequency thermal conductivity $\tilde{\kappa}(\omega)$ contains a weakly dispersive resonance at frequency $\omega_0 \sim \Gamma$, see Eqs. (52,53). This resonance is similar to the one found in Ref. [20] for parametric excitation of the SYK quantum dot; its quality factor is $Q \sim \ln\frac{\omega_0}{2\pi T}$. In general, our results are fully compatible with those obtained in Ref. [20]; the difference between Majorana and complex fermion models consists just in the presence of charge transport in the latter one.

Fig. 1 provides schematic view of the "phase diagram" of the model we studied, in the case $\Gamma \gg w$. We used logarithmic axis for the ratios $\omega/w$ and $T/w$ to make our phase diagram compact. Depending on relations between temperature $T$, frequency $\omega$ and transport bandwidth $w$, various transport regimes discussed above can found in the phase diagram; they are indicated in the Figure. 1 in the self-explanatory way; blue region in the centre corresponds to crossover between various specific regimes.

Recently we became aware of interesting preprint [21] discussing phenomenology of several strongly interacting fermionic systems, like overdoped $La_{2-x}Sr_xCuO_4$ and $^3He$ under

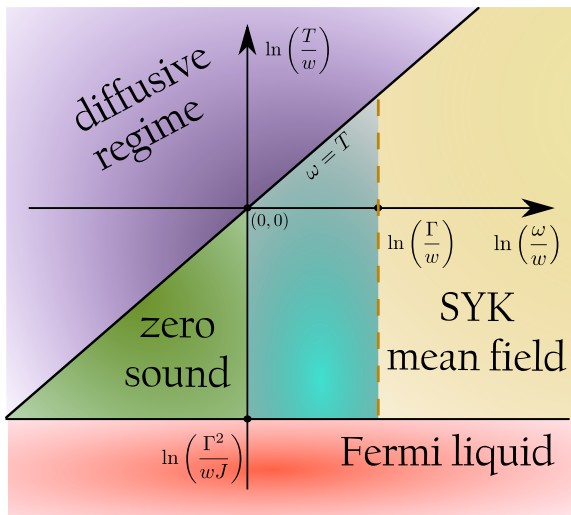

Figure 1: Transport regimes

strong pressure. One of important observations made in Ref. [21] is that both these very different objects demonstrate significant deviation from the "universal" Kadowaki-Woods ratio $A/\gamma^2$ known for many various electronic materials including strongly correlated ones; here $A$ is the coeffcient in the $T^2$ law of resistivity known for Fermi-liquids, $R(T) = R_0 + AT^2$, and $\gamma$ enters specific heat, $C(T) = \gamma T$. Namely, for $La_{1.67}Sr_{0.33}CuO_4$ the ratio $A/\gamma^2$ is anomalously large; for the compressed $^3He$ one cannot define conductivity, but the analysis of transport relaxation rate leads to very similar conclusion as for $La_{1.67}Sr_{0.33}CuO_4$.

Kadowaki-Woods ratio was discussed in Ref. [8] where it was found to be universal; however in the model [8] the only quadratic terms were the tunneling ones. Here we would like to notice that "heavy Fermi-liquid" with anomalously large Kadowaki-Woods ratio can be obtained within $SYK_4 + SYK_2$ model we consider here, if we assume $\Gamma \gg w$. Indeed, in this model conductivity scales [8] as $\sigma \sim \frac{w^2}{JT}$ at $T > T_{FL} = \Gamma^2/J$. Below $T_{FL}$ we expect Fermi-liquid behavior to set in, with $R(T) = AT^2$, thus by continuity arguments $A \approx (J/w\Gamma)^2$. Then, the specific heat coefficient $\gamma \approx J/\Gamma^2$, and the Kadowaki-Woods ratio is $A/\gamma^2 \sim \Gamma^2/w^2 \gg 1$ and does not depend on $J$. We conclude that the model studied in this paper may be relevant for the description of extremely strongly interacting Fermi systems, like those discussed in Ref. [21].

# Acknowledgements

We are grateful to Alexei Kitaev, Kamran Behnia and Grigory Tarnopolsky for useful discussions. We are also grateful to Aleksandr Svetogorov for his comments to the text.

**Funding information** Research of A.V.L. was partially supported by the Basis Foundation, by the Basic research program of the HSE and by the RFBR grant # 20-32-90057.

# A  Action and continuity equations in the presence of the sources

In this appendix we will cover gaps of the main text. Particular, we have not written action with sources to simplify formulas.

The action for fields $G$ and $\Sigma$ in the presence of sources has the following form:

$$
\begin{aligned}
S = &-iN \sum_{\mathbf{r}} \int_{\mathcal{C}} dt_1 dt_0 (\Sigma(t_1, t_0|\mathbf{r}) + \Sigma_{free}(t_1, t_0)) G(t_0, t_1|\mathbf{r}) - iN\ tr \log\{\hat{\Sigma}\} \\
&+ iN \int_{\mathcal{C}} dt_1 dt_0 \sum_{\mathbf{r}} \left\{ \frac{J^2}{4} [G(t_1, t_0|\mathbf{r})G(t_0, t_1|\mathbf{r})]^2 + \frac{\Gamma^2}{2} G(t_1, t_0|\mathbf{r})G(t_0, t_1|\mathbf{r}) \right\} \\
&\qquad\qquad\qquad\qquad\qquad\qquad\qquad \times (1 + L(\mathbf{r}, t_1) + L(\mathbf{r}, t_0)) \\
&+ iN w^2 \int_{\mathcal{C}} dt_1 dt_0 \sum_{\mathbf{r},\delta\mathbf{r}} G(t_1, t_0|\mathbf{r})G(t_0, t_1|\mathbf{r}+\delta\mathbf{r})\Big(1 + \frac{L(\mathbf{r}, t_1) + L(\mathbf{r}+\delta\mathbf{r}, t_1)}{2} \\
&\qquad\qquad\qquad\qquad\qquad\qquad\qquad + \frac{L(\mathbf{r}, t_0) + L(\mathbf{r}+\delta\mathbf{r}, t_0)}{2}\Big) \\
&+ \exp\{i\varphi^{(U)}(\mathbf{r}+\delta\mathbf{r}, t_1) - i\varphi^{(U)}(\mathbf{r}, t_1) - i\varphi^{(U)}(\mathbf{r}+\delta\mathbf{r}, t_0) + i\varphi^{(U)}(\mathbf{r}, t_0)\} \,.
\end{aligned}
\tag{55}
$$

Our aim will be to find the action for the soft modes in the presence of sources. Using this action we apply procedure 16 from the main text.

After substitution of the field $G$ from the soft manifold we can find that terms proportional to $\Gamma$ or $w$ contains logariphimcaly divergent integral (for $q = 4$). This devergency does not affect on the action of the soft mode but to find one we need to use the following regularization procedure. We should consider the problem with general $q$ and expand action up to the second order in soft mode fluctuation, after than we need to take the limit $q \to 4 - 0$. The similar procedure was also performed in [20]. As a result we have the quadratic action for soft mode, which can be written in the Fourier domain as:

$$
\begin{aligned}
S_{\text{final}} = &\frac{1}{2} \int \frac{d^d\mathbf{p}d\Omega}{(2\pi)^{d+1}} \left(\hat{\varphi}_{\mathbf{p},\Omega} - \mathcal{E}(\hat{f}_{\mathbf{p},\Omega} - \frac{i}{\Omega}\hat{L}_{\mathbf{p},\Omega})\right)^\dagger \left[\hat{\mathcal{G}}_0^{(\varphi)}(\Omega)\right]^{-1} \left(\hat{\varphi}_{\mathbf{p},\Omega} - \mathcal{E}(\hat{f}_{\mathbf{p},\Omega} - \frac{i}{\Omega}\hat{L}_{\mathbf{p},\Omega})\right) \\
&+ \frac{1}{2} \int \frac{d^d\mathbf{p}d\Omega}{(2\pi)^{d+1}} \left(\hat{f}_{\mathbf{p},\Omega} - \frac{i}{\Omega}\hat{L}_{\mathbf{p},\Omega}\right)^\dagger \left[\hat{\mathcal{G}}_0^{(f)}(\Omega)\right]^{-1} \left(\hat{f}_{\mathbf{p},\Omega} - \frac{i}{\Omega}\hat{L}_{\mathbf{p},\Omega}\right)^\dagger \\
&- \frac{1}{2} \int \frac{d^d\mathbf{p}d\Omega}{(2\pi)^{d+1}} \left\{\left(\hat{f}_{\mathbf{p},\Omega} - \frac{2i}{\Omega}\hat{L}_{\mathbf{p},\Omega}\right)^\dagger \left[\hat{\Sigma}^{(f)}(\Omega) + \hat{\Sigma}_w^{(f)}(\Omega)\right]\left(\hat{f}_{\mathbf{p},\Omega} - \frac{2i}{\Omega}\hat{L}_{\mathbf{p},\Omega}\right)\right. \\
&\left.\hspace{6cm} + \hat{f}_{\mathbf{p},\Omega}^\dagger \left[\hat{\Sigma}_w^{(f)}(\Omega,\mathbf{p}) - \hat{\Sigma}_w^{(f)}(\Omega,\mathbf{0})\right]\hat{f}_{\mathbf{p},\Omega}\right\} \\
&- \frac{1}{2} \int \frac{d^d\mathbf{p}d\Omega}{(2\pi)^{d+1}} \left\{\hat{L}_{\mathbf{p},\Omega}^\dagger \hat{\Sigma}_t^{(L,f)}(\Omega,\mathbf{p})\hat{f}_{\mathbf{p},\Omega} + \hat{f}_{\mathbf{p},\Omega}^\dagger \hat{\Sigma}_t^{(f,L)}(\Omega,\mathbf{p})\hat{L}_{\mathbf{p},\Omega}\right\} \\
&- \frac{1}{2} \int \frac{d^d\mathbf{p}d\Omega}{(2\pi)^{d+1}} \left(\hat{\varphi}_{\mathbf{p},\Omega} - \mathcal{E}\hat{f}_{\mathbf{p},\Omega} + \hat{\varphi}_{\mathbf{p},\Omega}^{(U)}\right)^\dagger \hat{\Sigma}_t^{(\varphi)}(\Omega,\mathbf{p})\left(\hat{\varphi}_{\mathbf{p},\Omega} - \mathcal{E}\hat{f}_{\mathbf{p},\Omega} + \hat{\varphi}_{\mathbf{p},\Omega}^{(U)}\right).
\end{aligned}
\tag{56}
$$

This action without sources was written in the main text as: (19). Besides operators $\mathcal{G}$ and $\hat{\Sigma}$ mentioned in the main text we also introduced the following operators:

$$
\Sigma_{t,(R)}^{(f,L)}(\Omega,\mathbf{p}) = -\Sigma_{t,(R)}^{(L,f)}(\Omega,\mathbf{p}) = C_w \sum_{\delta\mathbf{r}} i\Omega \left(\frac{\mathbf{p}\delta\mathbf{r}}{2}\right)^2 \psi(\Omega).
\tag{57}
$$

The variation of the above action leads us to continuity equations (see (5.1,5.2)) which has the following form in the presence of sources:

$$
0 = -\frac{\delta S_{\text{final}}}{\delta\varphi_{\mathbf{p},\Omega}^{(q)\dagger}} = -iC_Q\Omega\left(V_{\mathbf{p},\Omega} + \mathcal{E}L_{\mathbf{p},\Omega}\right) + 8\tilde{\sigma}p^2\frac{\tilde{\psi}(\Omega)}{-i\Omega}\left(V_{\mathbf{p},\Omega} + U_{\mathbf{p},\Omega}\right),
$$

$$
0 = -\frac{\delta S_{\text{final}}}{\delta f_{\mathbf{p},\Omega}^{(q)\dagger}} = \mathcal{E}\frac{\delta S_{\text{final}}}{\delta\varphi_{\mathbf{p},\Omega}^{(q)\dagger}} - iC_E\Omega(\Omega^2 + 1)\left(\tau_{\mathbf{p},\Omega} - L_{\mathbf{p},\Omega}\right)
\tag{58}
$$

$$
+ \frac{C_\Gamma}{2}i\Omega\psi(\Omega)\left(\tau_{\mathbf{p},\Omega} - 2L_{\mathbf{p},\Omega}\right) + \frac{\tilde{\sigma}}{4}p^2\left(\left((\Omega^2+1)\frac{\psi(\tilde{\Omega})}{-i\Omega} + i\Omega\psi(\Omega)\right)\tau_{\mathbf{p},\Omega} + i4\Omega\psi(\Omega)L_{\mathbf{p},\Omega}\right).
$$

These equations describes charge and energy conservations respectively. Here we also introduced notations:

$$
V_{\mathbf{p},\Omega} \equiv -i\Omega\left(\varphi_{\mathbf{p},\Omega}^{(cl)} - \mathcal{E}f_{\mathbf{p},\Omega}^{(cl)}\right), \qquad \tau_{\mathbf{p},\Omega} \equiv -i\Omega f_{\mathbf{p},\Omega}^{(cl)}.
\tag{59}
$$

We need to solve continuity equations to find changes of fields $\varphi$ and $f$ caused by sources. Using this information we can find currents using (34). It leads us to the expression (35) which is the main result of the text.

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
