# Peer review of "High-frequency transport and zero-sound in an array of SYK quantum dots"

_SciPost Physics, doi:SciPost Phys. 13, 073 (2022)_

## Round 1 · Referee Report · Anonymous (Referee 1) · 2022-1-10

Report

The manuscript is devoted to finite frequency and momentum response function of a model based on SYK array. The authors derived dynamic response functions and found an analog of zero-sound mode. This is probably an interesting contribution, which might deserve publication. Unfortunately, it is really hard to judge more definitively due to poor quality of presentation.

Indeed the manuscript is much preoccupied with the technicalities (which are yet
notationally so obscure that they hardly help to understand anything) in expense of discussing the physics of the model. I would urge the authors to significantly expand introduction and/or conclusion sections to explain the assumptions, mechanics and consequences of their results. Some of the items to cover include:
(i) Phase diagram (e.g. energy vs momentum) which demonstrates hierarchy of
energy/momentum scales and expected window of existence of the zero sound.
(ii) Discussion of a usual hydrodynamic sound, its range of existence and overlap (if any) with the zero-sound.
(iii) Since the model is intrinsically disordered, one could expect a range where response functions exhibit diffusive behavior. Is it indeed the case and how does it conform to the presented expressions?
(iv) What are the relative contributions of reparametrization and phase fluctuations in the obtained response functions. E.g. how different are they in
Majorana SYK (no phase fluctuations), vs. Fermi liquid (no reparametrizations)?
(v) What is the role of finite N (if any) for the presented results?

The list can go on and the authors are surely aware of other potential questions, which should be addressed. I want to encourage them not to be shy to provide much more context to their findings.

---

## Round 1 · Referee Report · Anonymous (Referee 2) · 2022-2-4

Strengths

The results obtained in this work are original and improve our understanding of the large spectrum of one-dimensional systems that can be constructed from SYK-like quantum dots.

The technical presentation is detailed, allowing the reader to follow and reproduce most of the computations, if desired.

The scope of the paper is clearly defined, and the authors do not digress or speculate beyond it.

Weaknesses

The discussion of the physical interpretation and relevance of the results is mostly neglected in favor of the technical details.

The notation used in this work can feel unfamiliar to readers not sharing a common background with the authors, and may be excessive given the simplicity of the underlying system under consideration.

Given that one of the strong points of SYK-like models is our ability to solve them exactly at large $N$, it is a pity that nowhere in this work numerical solutions are used to complement or at least support the analytical arguments. This would greatly improve the presentation, allowing the reader to more easily visualize the results obtained in the various regimes under consideration.

Report

Building on and extending previous work by the same authors, this paper aims to compute the conductivity properties of a one-dimensional array of $q=4$ random-mass complex SYK systems, coupled by quadratic terms of random tunneling between neighboring sites. Expressions for the electric and thermal conductivities are obtained in the large-$N$ limit as functions of the frequency and momentum, to be later expanded in the high- and low-frequency regimes. A non-universal Lorentz ratio is found in the latter regime, as well as a zero-sound mode in the electric conductivity at high frequencies.

Given that zero-sound is traditionally associated to Fermi-liquids, it is certainly puzzling to observe this phenomenon precisely in an SYK-like system, i.e. a model of non-Fermi liquids. The authors, however, decide not to dwell on this matter and instead concentrate on the technical aspects. This referee feels that the presentation would greatly benefit from moving some of the computational details to an appendix, while devoting more effort to discuss the physics involved, as has already been suggested by another report.

Requested changes

1- A considerably improved introduction should be provided discussing the larger context of the model, the physical interpretation and consequences of zero-sound, etc. The conclusions should also be extended, since at this time they amount to a summary of results presenting no insight into their relevance, connections to other known systems, and so on.

2- At the end of section 3.2 the authors discuss the asymmetry parameter $\mathcal{E}$, mentioning numerical evidence that its absolute value is found to be bounded. The provided citations, however, deal with a different variant of the complex SYK model, namely the mass-deformed case where the coefficient in front of the quadratic term in each quantum dot is not random but simply a fixed constant. This is different from the ${\rm SYK}{q=4} + {\rm SYK}$ considered here, and can lead to confusion. To avoid this, the authors should clarify the distinctions between ${\rm SYK}{q=4} + {\rm SYK}$ and the mass-deformed SYK model, and maybe comment on whether their results would also apply to a linear array of mass-deformed complex SYK quantum dots.

3- In equation (12) the authors introduce the parameter $q$ because it "will be useful below for dimensional regularization of some singular expressions." Details of this regularization are scarce to non-existent, and the reader is left wondering whether the results obtained could depend on this regularization scheme. Some discussion of this procedure is therefore in order.

4- While the manuscript is readable as is, the number of typographical errors can be distracting, and there is ample room for improving the English.

---

## Round 2 · Referee Report · Anonymous (Referee 2) · 2022-6-21

Report

The authors have addressed in this new version all issues that were raised about their original submission, thus I recommend this revised manuscript be published.

---

## Round 2 · Referee Report · Anonymous (Referee 3) · 2022-8-9

Strengths

1 - clear structure
2 - interesting and novel results
3 - deep analytical study
4 - clear connection with what was made before (on a technical level)

Weaknesses

1 - the physical motivation of the research was not very well explained
2 - absence of numeracal cross-check

Report

I would recommend that the paper is accepted after the minor revisions according to the comments presented in this review.

Requested changes

listed in the report

Attachment

---

## Round 2 · Author Response

Dear Editors,

We thank both referees for their reviews of our work. We took these reports into account while doing a major revision to our work. We have extended both the Introduction and Conclusions. We also simplified formulas for a more convenient presentation and added an Appendix for the readers interested in technical details. Below we answer the referees’ comments in detail.

Answers to the Report 1 on 2022-1-10

(i) Phase diagram (e.g. energy vs momentum) which demonstrates the hierarchy of energy/momentum scales and the expected window of the existence of the zero sound. We have added a phase diagram to describe regions with different features to the Conclusions (ii) Discussion of a usual hydrodynamic sound, its range of existence, and overlap (if any) with the zero-sound. The presented model lives on the lattice. The tunneling elements between dots do not depend on the lattice constants. Therefore, there is no traditional sound in the model. We have called the mode a zero-sound for the following reasons: - This mode describes the fluctuation of electron density - This mode exists only for high frequencies which are much higher than inverse thermalization time

(iii) Since the model is intrinsically disordered, one could expect a range where response functions exhibit diffusive behavior. Is it indeed the case and how does it conform to the presented expressions? The diffusive regime has detailed analysis in our paper (see section 5.3 of the new text, it was also present in the old version in section 5.2)

(iv) What are the relative contributions of reparametrization and phase fluctuations in the obtained response functions. E.g. how different are they in Majorana SYK (no phase fluctuations), vs. Fermi liquid (no reparametrizations)? We did not describe a crossover to Fermi liquid so our formulas do not cover this region of parameters. The comparison between the Fermi-liquid theory and our result is presented in the conclusion. We also mentioned in both versions of the text that in the absence of the charge transport there is only an intrinsic contribution to the heat conductivity.

(v) What is the role of finite N (if any) for the presented results? Finite N bounds the strength of \Gamma where our approach is valid. For \Gamma less than \Gamma_c ~ J/N, we can no longer consider fluctuations of the soft modes as Gaussian. The corresponding comment is added to the 1st paragraph of the Conclusions.

Answers to the Report 2 on 2022-2-4

  1. A considerably improved introduction should be provided discussing the larger context of the model, the physical interpretation and consequences of zero-sound, etc. The conclusions should also be extended, since at this time they amount to a summary of results presenting no insight into their relevance, connections to other known systems, and so on. We have improved our Introduction and Conclusion.

  2. At the end of section 3.2 the authors discuss the asymmetry parameter E, mentioning numerical evidence that its absolute value is found to be bounded. The provided citations, however, deal with a different variant of the complex SYK model, namely the mass-deformed case where the coefficient in front of the quadratic term in each quantum dot is not random but simply a fixed constant. This is different from the SYKq=4+SYKq=2 considered here, and can lead to confusion. To avoid this, the authors should clarify the distinctions between SYKq=4+SYKq=2 and the mass-deformed SYK model, and maybe comment on whether their results would also apply to a linear array of mass-deformed complex SYK quantum dots. In section 3.2 we are considering the saddle-point equations of the pure SYK model with real fermions. The authors of the mentioned papers considered different models but obtain the same saddle-point equations as ours. As a result, their numerical analysis of these equations is applicable to our case.

  3. In equation (12) the authors introduce the parameter q because it "will be useful below for dimensional regularization of some singular expressions." Details of this regularization are scarce to non-existent, and the reader is left wondering whether the results obtained could depend on this regularization scheme. Some discussion of this procedure is therefore in order. We add the comment in the Appendix between (54) and (55)
  4. While the manuscript is readable as is, the number of typographical errors can be distracting, and there is ample room for improving the English. Hopefully, the number of typographical errors and cases of improper use of English is now reduced considerably.

---

## Round 2 · List of Changes

1. Introduction and conclusion were expanded.
  2. Appendix was added.
  3. The phase diagram with different transport regimes was added
  4. The section " Noether’s theorem" was added
  5. Formulas were simplified in the main part of the text
  6. Large number of small correction was made

---

## Round 3 · Author Response

Dear Editors,
please find the updated version of our manuscript, with comments from the 2nd Referee taken into account. We have performed the requested small changes: the abstract and introduction were extended, and typos were corrected.
Below we would like to answer point-by-point the second referee report which includes 11 comments.
- Introduction and abstract were extended (see the end of the first page and beginning of the second page for changes in the Introduction).
- the sentence where Jakiw-Teitelboim gravitation was mentioned is now removed
- We added information about J at the beginning of the Introduction
- The sentence mentioned by the Referee was corrected
- We added a statement on the symmetric properties of the tensors
- In general, it is possible to write the mentioned sum with only one term but it leads to the additional symmetry for tunneling amplitude. We decided to write two terms but without any symmetry limitations on t.
- We removed sources from eq.(5)
- We added chemical potential to the Hamiltonian
- We added the reference to the paper where the solution (11-12) was provided originally.
- We fixed equations in accordance with the Referee's comment.
- We do not agree with Referee's statement that "absence of numerical cross-check" constitutes a weakness of our present research. All calculations described in this manuscript are parametrically exact. We did not employ any uncontrolled approximations or hypotheses. Therefore there is no reason to "support" these results by numerics.
All mentioned typos were fixed.
We believe that now our manuscript is ready for publication.
Sincerely yours,
Aleksey Lunkin and Mikhail Feigel'man

---

## Editorial Decision

published